# Searching for γ-ray Emission from Binary Black-Hole Mergers Detected in LIGO/Virgo O3 Run

Chongyang Ren [1,†] and Zhongxiang Wang [1,2,*,†]

1    Department of Astronomy, School of Physics and Astronomy, Yunnan University, Kunming 650091, China
2    Key Laboratory for Research in Galaxies and Cosmology, Shanghai Astronomical Observatory, Chinese Academy of Sciences, 80 Nandan Road, Shanghai 200030, China
*    Correspondence: wangzx20@ynu.edu.cn
†    These authors contributed equally to this work.

**Abstract:** We conduct searches for γ-ray emission from the binary black-hole (BBH) mergers reported in the Gravitational-Wave Candidate Event Database (GraceDB). The γ-ray data are from the all-sky survey of the Large-Area Telescope (LAT) onboard the *Fermi Gamma-ray Space Telescope (Fermi)*, which allows searches for events of given time durations in large sky areas. The Two-Micron All-Sky Survey Photometric Redshift Catalog (2MPZ) is used for target selection, from which galaxy sources within the 90% credible areas and the distance ranges given by the gravitational-wave (GW) detections are determined. Excluding those BBH cases with large credible areas and/or dense fields (containing too many 2MPZ sources), searches for short transient events over the time duration of from $-1$ to 100 days of a trigger time for seven BBH mergers are conducted. We find two candidate short flaring events in the field of the GW event S200311bg and one in that of S190408an. However, the flaring events all have low significance (after considering the trial factor), and the third one appears off the position of the target galaxy. We discuss one of them from the first field that is detected in different short time-bin data and suggest that it is possibly a real flare arising from a radio galaxy. More such studies for the near-future GW detection run are planned, for which we will adjust our search strategy to be more effective and target flares of various time scales.

**Keywords:** gravitational waves; transients; black hole mergers; gamma-rays; stars

## 1. Introduction

The first direct detection of gravitational waves (GWs) was from the merger of two black holes (BHs) that had masses of $\sim$36 $M_\odot$ and $\sim$29 $M_\odot$ [1]. This detection (named GW150914) with the Laser Interferometer Gravitational-Wave Observatory (LIGO) not only marks the starting point of the new era for astrophysical studies, i.e., the era of multi-messenger astrophysics (e.g., [2]), but also hints at many such GW events due to binary BH (BBH) mergers [3]. Indeed, after it was followed with the second reported case, GW151226, the merger of two stellar-mass BHs [4], 10 BBH mergers (with the most-known binary neutron star merger GW170817 as an additional case; [5]) were reported to be detected from the nearly two years of LIGO and Virgo observation runs O1 and O2 [6]. Then, in a nearly one-year O3 run, the dominant GW events, with more than 70 cases reported, are BBH mergers [7,8].

Taking GW150914 as an example, most of the BBH mergers were estimated to contain relatively heavy BHs, i.e., having masses greater than those ($<$15 $M_\odot$) of the BHs [9] identified in Galactic X-ray binaries through electromagnetic (EM) wave observations. The finding raises questions such as why these heavy BHs were previously unknown from astronomical EM studies and also how such BBH systems are formed. Extensive theoretical studies for understanding the formation processes of these BBH systems have been carried out [10]. Two physically-different types of scenarios are under consideration: analogous to





the formation of Galactic stellar-mass BH X-ray binaries [11] and dynamical interactions in high stellar-density regions such as star clusters and nuclear clusters. In order to establish a clear scenario for the formation of the GW-event BBHs, on one side, searches and studies of possible progenitor systems should be conducted (e.g., [12]), and on the other side, localizations of the GW events are particularly needed such that their locations in galaxies would be known: are they at the center, the galactic disk, or the edge?

While the mergers of binary neutron star (NS) systems certainly emit EM waves (cf., the remarkable GW170817; [2]), whether BBH (or BH+NS) mergers would be able to produce observable radiation is under both extensive theoretical and observational investigations. Triggered by the tentative detection of a short $\gamma$-ray burst (GRB) associated with GW150914 [13,14], which was questioned by other analysis of the same data [15], a variety of scenarios was proposed for possible physical processes that might induce EM emission (e.g., [16–21]). Among the possibilities, a general consideration along the thinking line of activities of accreting BHs is the existence of material of remnant disks or circum-BHs. Thus types of GRB-like prompt emission, radio emission from jets launched from BHs, and cosmic rays and high-energy emission have been suggested to be able to arise from BH merger events. In addition, accretion disks of the central supermassive BHs of galaxies could be the preferable sites for BBH mergers, and the mergers could induce gas interactions and ultraviolet/optical emission (e.g., [22]). The candidate optical flare from an active galactic nucleus at $\sim$34 days after the BBH merger event S190521g was recently reported [23], seemingly in support of the latter scenario.

Numerous searches for EM signals associated with BBH mergers have been conducted. These searches include prompt observations following GW candidate event alerts distributed through the Gamma-ray Coordinates Network (GCN), which have been collectively posted at the Gravitational-Wave Candidate Event Database (GraceDB)[1]. Follow-up (targeted) searches have also been conducted, mostly through a survey mode ( to cover the large credible areas of the GW events). For example, there was the intermediate Palomar Transient Factory survey at optical bands [24], with which the Karl G. Jansky Very-Large Array followed in coordination at radio frequencies [25]. Further, there were radio searches utilizing the large field of view (FOV) provided by the Australian Square Kilometre Array Pathfinder (e.g., [26]) or the Low-Frequency Array (e.g., [27]). At high energies and very-high energies, in addition to the routine searches for GRB-like events, there were X-ray searches with the *Gehrels Swift Observatory* [28] and $\gamma$-ray searches with the AGILE satellite [29] and the High-Energy Stereoscopic System (H.E.S.S.; [30]).

When searching for possible EM signals associated with the GW events, the technical challenges are their large, credible regions ($\sim$100 deg$^2$) and uncertain time ranges requiring observational coverage. For this reason, large-FOV survey telescopes are the optimal facilities for carrying out searches. Given this, and encouraged by the multi-wavelength effort such as the candidate S190521g detection at optical bands, we conducted searches for high-energy $\gamma$-ray emissions from GW events by taking advantage of the nearly all-time and all-sky coverage capability of the Large-Area Telescope (LAT; [31]) onboard the *Fermi Gamma-ray Space Telescope (Fermi)*. In this work, we focused on the candidate BBH merger events in the LIGO/Virgo O3 run recorded in GraceDB, which were the dominant ones among the GW events. Following each of the reported GW events, searches for EM signals with *Fermi* Gamma-ray Burst Monitor at hard X-rays and soft $\gamma$-rays and *Fermi* LAT within short time periods of $\sim$10 ks of a trigger were routinely conducted (see related reports on GraceDB). Our work extended the time periods to 100 days post an event and may be considered as searches for "delayed EM signals." Below we first describe our selection of the BBH merger targets and the searching method in Section 2 and the *Fermi* LAT data and analysis in Section 3. The results are presented in Section 4. We discuss the results in Section 5.

## 2. Selection of BBH Merger Targets and Searching Method

The LIGO/Virgo O3 run went through the periods of 1 April 2019 to 30 September 2019, the first half (O3a), and from 1 November 2019 to 27 March 2020, the second half (O3b). In this run, the GW triggers were publicly announced in real time for the first time (for more details, see https://gracedb.ligo.org/superevents/public/O3/ (accessed on 29 September 2022)). We selected our targets based on public alerts. Among 56 announced detection candidates, 37 were classified as BBH mergers with high probabilities.

We then further selected those of the BBH mergers with probabilities of $\geq 90\%$ and 90% credible areas $\leq 300\,\mathrm{deg}^2$. The latter requirement was to avoid too large sky areas and thus too many sources to be analyzed. This selection excluded 25 out of 37 reported BBH mergers. GraceDB provides a skymap for each of the mergers and an ellipse of a 90% credible area for each of a few with positions relatively well determined. In such a given area, source positions were needed in order to search for possible $\gamma$-ray emissions by analyzing *Fermi* LAT data. We used the Two-Micron All-Sky Survey (2MASS) Photometric Redshift Catalog (2MPZ; [32]) for selecting sources that contain one million galaxies with estimated redshift information. The galaxy sources in 2MPZ are located within a 90% credible area of a merger event, provided by either an ellipse region or a skymap, and were thus selected. As the redshift range (or the source distance range) of a merger is also given, the 2MPZ sources were further selected by matching their redshifts with the given redshift range. We noted that the redshifts in 2MPZ extend to 0.3, with an accuracy of 0.015 ($1\sigma$), which approximately cover the redshift ranges of the BBH merger events. In this step of the target source selection, the Planck Cosmological parameters, $H_0 = 67.4\,\mathrm{km\,s^{-1}\,Mpc^{-1}}$ and $\Omega_m = 0.315$ [33], were used to convert redshifts to luminosity distances.

However, for four and one merger events, the 2MPZ sources in each of the given credible areas plus redshift ranges were too dense or zero, respectively. In the first cases, the average galaxy density was $>2\,\mathrm{deg}^{-2}$, which would cause confusion in the *Fermi* LAT data analysis (note that the angular resolution of LAT can be as large as several degrees depending on photon energies[2]). These five merger events were further excluded. In the end, seven BBH mergers were chosen to be studied. In Table 1, information on them is provided. The numbers of galaxies found in 2MPZ for each merger event that match the 90% credible areas and the redshift ranges are listed in the table. Transient $\gamma$-ray emissions from these galaxies were searched in *Fermi* LAT data. Taking the candidate optical flare found for S190521g [23] as a reference example, we enlarged their searching range of $\leq 60$ days after a merger-event trigger to a time period of from $-1$ to $100$ days.

**Table 1.** Information for seven BBH merger events, where column 6 provides the number of galaxy targets found in 2MPZ.

| Event ID | Source (Probability) | Location | 90% Credible Area (deg$^2$) | Luminosity Distance (Mpc) | Number |
|---|---|---|---|---|---|
| S200311bg | BBH (>99%) | ellipse (00$h$09$m$, $-$08$d$14$m$, 6.91$d$, 1.59$d$, 72.77$d$) | 34 | $1115 \pm 175$ | 31 |
| S200225q | BBH (96%), Terrestrial (4%) | ellipse (08$h$28$m$26$s$, $+$87$d$45$m$51$s$, 3.38$d$, 2.54$d$, 104.41$d$) | 27 | $995 \pm 188$ | 1 |
| S200224ca | BBH (>99%) | ellipse (11$h$38$m$42$s$, $-$09$d$20$m$21$s$, 9$d$, 3$d$, 68$d$) | 73 | $1585 \pm 331$ | 13 |

**Table 1.** *Cont.*

| Event ID | Source (Probability) | Location | 90% Credible Area (deg$^2$) | Luminosity Distance (Mpc) | Number |
|---|---|---|---|---|---|
| S190701ah | BBH (93%), Terrestrial (7%) | ellipse (02$h$30$m$42$s$, −06$d$53$m$30$s$, 8$d$, 3$d$, 97$d$) | 67 | 1045 ± 234 | 2 |
| S190512at | BBH (95%), MassGap (5%) | GWTC-2 skymap | 226 | 1462 ± 347 | 35 |
| S190503bf | BBH (96%), MassGap (3%) | GWTC-2 skymap | 94 | 1527 ± 411 | 31 |
| S190408an | BBH (>99%) | GWTC-2 skymap | 139 | 1548 ± 302 | 14 |

## 3. *Fermi* LAT Data and Analysis

LAT onboard *Fermi* is a $\gamma$-ray imaging instrument scanning the whole sky every three hours in the energy range of from 50 MeV to 1 TeV [31,34]. For each galaxy target found for a merger event, we selected the P8R3_SOURCE class *Front+Back* events (i.e., `evclass = 128` and `evtype = 3`) in the energy range of 0.3–500 GeV. The low-energy 0.3 GeV was chosen because the instrument response function of LAT has relatively large uncertainties in the <0.3 GeV energy range, and also, the large point-spread function (>2 deg) may easily cause spurious detections (particularly when the detection-significance criterion is not high) due to the proximity to bright sources. The region of interest (RoI) was set to a 15-deg radius circle centered at the galaxy's position. The time period of the data considered was from −1 to 100 days of a merger trigger. We used `Fermitools 2.0.19`. The LAT events were reduced by requiring the zenith angle < 90°, DATA_QUAL >0, and LAT_CONFIG = 1. We noted that since 2018 March, no Target of Opportunity (ToO) observations have been performed with *Fermi*, and thus no additional ToO data were available for our targets.

A model file was constructed for this target based on the *Fermi* LAT fourth source catalog data release 2 (4FGL-DR2; [35]) by running `make4FGLxml.py`, for which the spectral parameters of all known 4FGL sources in the RoI were included. The Galactic background and extragalactic diffuse emission models, gll_iem_v07.fits and iso_P8R3_SOURCE_V3_v1.txt, respectively, were also included in the source model. The target was assumed to have a simple power law, $dN/dE = N_0(E/E_0)^\Gamma$ (where $\Gamma$ is the photon index), in the model file.

For this target, the unbinned maximum likelihood analysis was performed on the LAT data using the instrument response files (P8R3_SOURCE_V3) and the built source model. The time period selected was −1 to 100 days of the event trigger, while we set the data time-bins to be 1, 4, and 16 days (in other words, light curves of the target of these time-bins were constructed). In the analysis, for the 4FGL sources outside of or within 5 degrees of the target, their spectral parameters were fixed at the values given in 4FGL or set free, respectively. In addition, the normalizations of the two background components were set as free parameters. The analysis was repeated for all the galaxy targets (127, as listed in Table 1). Because we were performing the analysis to the data that most likely contained no emission from our targets, we met non-convergence cases in our analysis. For such cases, if after either repeating the process or adjusting parameter ranges without successfully overcoming the problem, we assign them to have TS = 0.

We considered a possible detection when the test statistic (TS) value of a light-curve data point was ≥9 (i.e., ≥3$\sigma$ detection). For the galaxy targets in the fields of three BBH mergers, no TS ≥ 9 data points were found. The other four contained candidate detections. We then calculated a TS map of a size of 5° × 5° for each candidate by running `gttsmap`. The TS map was used to check if the TS ≥ 9 value at a target's position was caused by a point-source-like detection and not by nearby sources or extended structures in the field. We were able to exclude two candidates. Specifically, in the field of S190512at, we found that the known *Fermi* LAT source 4FGL J1512.8−0906 had a flare occurring around

MJD 58617–58621, and relatively high TS data points for a nearby galaxy target were induced by the flare.

## 4. Results

In our analysis, we found three possible short flaring events, with each arising from a galaxy in the fields of two merger events, S200311bg and S190408an. Two were from the field of the first and one from that of the second. For each of the candidate flaring events, we checked the positions of the Sun and the Moon at the determined flaring times, as they might affect the detections (e.g., [36]). They were not close to the respective RoIs. We also checked if there were extended sources in the RoIs and found in the second merger event only that there were two catalogued, extended sources, FHES J2304.0+5406 and CTB 109. However, they were ~9 and ~14 deg away with circular radii of ~1.6 and 0.26 deg, respectively, likely not affecting our results. In the following sub-sections, the detection results are presented.

### 4.1. S200311bg

The trigger time of this event was 11 March 2020 11:58:53.398 (UTC), or ~MJD 58919. In its 90% credible area ($\simeq$34 deg$^2$) and distance range ($1115 \pm 175$ Mpc), there are 31 2MPZ sources (cf., Table 1). We analyzed the data for the sources and found two short flaring events from two sources. Table 2 provides the information for the possible detections.

**Table 2.** Information for two candidate short flaring events in the field of S200311bg, where the serial number for a source is the sequence number in 2MPZ.

| Source (Serial Number) | RA (deg) | DEC (deg) | $z$ | Bin (day) | Time (MJD) | TS |
|---|---|---|---|---|---|---|
| J0004−0825 (71482) | 1.0036 | −8.4207 | 0.22 | 1 | 59007 | 11 |
| | | | | 1 | | 15 |
| J0006−1030 (504460) | 1.5313 | −10.5091 | 0.21 | 4 | 58950 | 12 |
| | | | | 16 | | 10 |

The first was from J000400.89−082515.8, at a time of 88 days after the trigger. The TS value is low, ~11, while the TS map shows a point-source-like detection, not due to some extended structures while often appearing in regions of the Galactic plane (see Figure 1). We ran `gtfindsrc` in `Fermitols` to determine the error circle of the $\gamma$-ray source, and the galaxy target is within the 2-$\sigma$ error circle. For this source, the obtained photon index was $\Gamma = 1.13 \pm 0.25$, and the photon flux was $1.0 \pm 0.7 \times 10^{-8}$ photon cm$^{-2}$ $^{-1}$ (corresponding to an energy flux of ~$4.7 \times 10^{-10}$ erg cm$^{-2}$ s$^{-1}$). The uncertainty of the flux was rather large. We thus tested to obtain the results in a higher energy range, 0.5–500 GeV, by excluding more low-energy photons from the background and nearby sources. The results were similar ($\Gamma \simeq 1.10$, TS $\simeq 12.6$), and the uncertainty was reduced to ~60%.

The second source is J000607.47−103032.8, detected at a time 31 days after the trigger. In the 1-day binned data analysis, the TS value was ~15, and this candidate detection was repeatedly found in the 4- and 16-day binned data analyses, with the TS values being ~12 and ~10, respectively. The TS maps from the three sets of the time-bin data were obtained, which verified the point-source detection was not caused by contamination from nearby sources (see Figure 2). We also determined its error circle and the target galaxy located within it. Each obtained photon index was $1.38 \pm 0.44$, $1.35 \pm 0.46$, or $1.65 \pm 0.49$ from 1-, 4-, or 16-day binned data, respectively, consistent with each other, but the fluxes had large, nearly 100% uncertainties. We found that when the energy range of 1.0–500 GeV was considered, the uncertainties could be reduced to ~50%. The fluxes (0.3–500 GeV) were $3.2 \times 10^{-9}$, $2.8 \times 10^{-10}$, and $6.3 \times 10^{-11}$ erg cm$^{-2}$ s$^{-1}$, respectively.

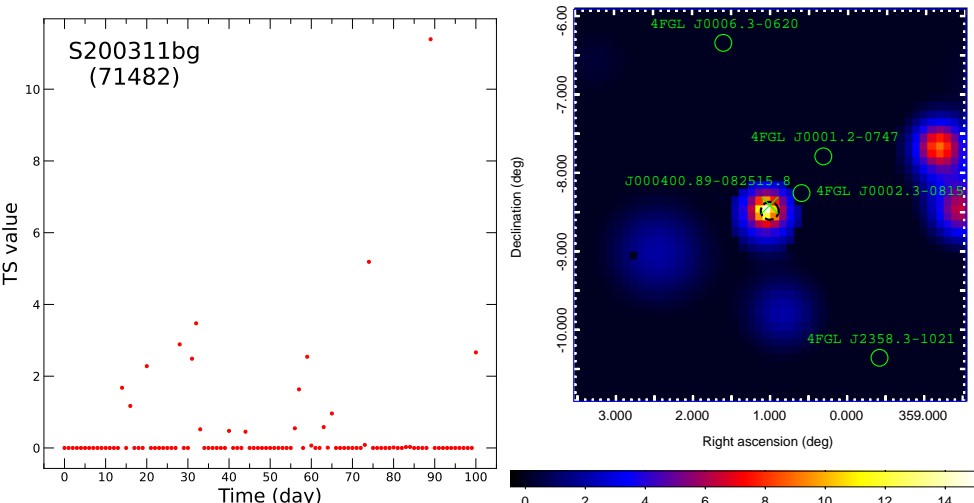

**Figure 1.** (**Left**) TS curve with 1-day time bin from −1 to 100 days of the BBH merger trigger S200311bg for J000400.89−082515.8 (71482). A TS ≃ 11 data point is seen at 88 days. (**Right**) 0.3–500 GeV TS map with a size of 5° × 5° for the high-TS data point. The 4FGL sources in the field are marked with circles. The position of the target galaxy is marked with a cross, while a dashed black circle is the 2-$\sigma$ error circle (radius 0.11 deg) determined for the candidate $\gamma$-ray source.

For these two flaring cases, we checked the events causing them. The primary ones for the first and second were 11.4 and 15.7 GeV photons, respectively. They were found from running `gtsrcprob` in `Fermitools` to have high probabilities of 99.997% and 90.4%, respectively, coming from each of the $\gamma$-ray sources. We also checked the variabilities of the nearby sources (shown in Figures 1 and 2). Among them, 4FGL J0001.2−0747 and 4FGL J2358.3−1021 were identified with `Variability_Index` >24, i.e., having a ≤1% probability of being a steady source. However, the photons had very low probabilities ($<2 \times 10^{-5}$) coming from them or other nearby steady sources. We also calculated TS maps without the field sources being removed. The two high `Variability_Index` sources and other sources all did not appear, the same as those shown in Figures 1 and 2 (note this is due to the short time bin). Thus the events were not likely caused by nearby sources.

### 4.2. S190408an

For this second BBH merger case, which had a trigger time of 2019-04-08 18:18:27 (UTC), or ∼MJD 58581, we also detected a candidate event among 14 2MPZ sources within the 90% credible area and the distance range. The information for this candidate detection is provided in Table 3. In the 1-day binned data analysis for the source J225050.22+452513.0, there was a data point at the 86-day delay time found with TS ≃ 10 (see Figure 3). The TS map obtained shows that the galaxy target is slightly off the $\gamma$-ray source, but the determined 2-$\sigma$ error circle still enclosed the galaxy. The obtained photon index for the candidate event was 2.51 ± 0.73, and the photon flux in 0.3–500 GeV was $4.8 \pm 2.8 \times 10^{-8}$ photon cm$^{-2}$ s$^{-1}$, corresponding to a flux of ∼$6.6 \times 10^{-11}$ erg cm$^{-2}$ s$^{-1}$.

The same as the above in Section 4.1, we checked any particular events causing this flaring case but did not find any high-energy photons. Further, no nearby sources shown in Figure 3 were identified with high `Variability_Index` values.

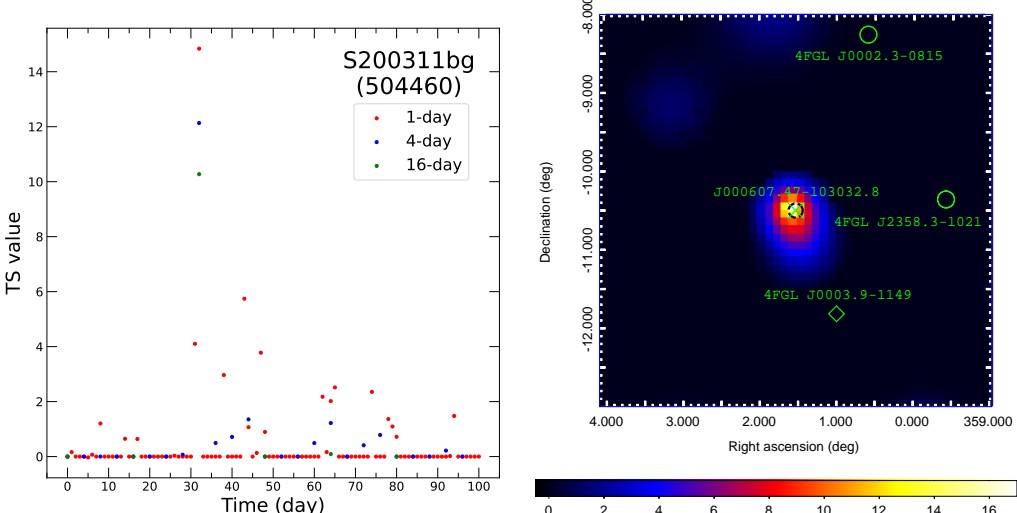

**Figure 2.** (**Left**) The 1-, 4-, 16-day binned TS curves from −1 to 100 days of the BBH merger trigger S200311bg obtained for J000607.47−103032.8 (504460). A TS≥ 9 event is seen at 31 days in all the TS curves. (**Right**) The 0.3–500 GeV TS map with a size of 5° × 5° made from the 1-day bin data that corresponds to TS ≃ 15. The 4FGL sources in the field are marked with circles or a diamond, the latter indicating a Log-Parabola spectral form for the source. The position of the target galaxy is marked with a cross, while the dashed black circle is the 2-$\sigma$ error circle (radius 0.09 deg) determined for the candidate $\gamma$-ray source.

**Table 3.** Information for a candidate short flaring event in the field of S190408an.

| Source (Serial Number) | RA (deg) | DEC (deg) | z | Bin (Day) | Time (MJD) | TS |
|---|---|---|---|---|---|---|
| J2250 +4525 (279555) | 342.7093 | 45.4203 | 0.25 | 1 | 58667 | 10 |

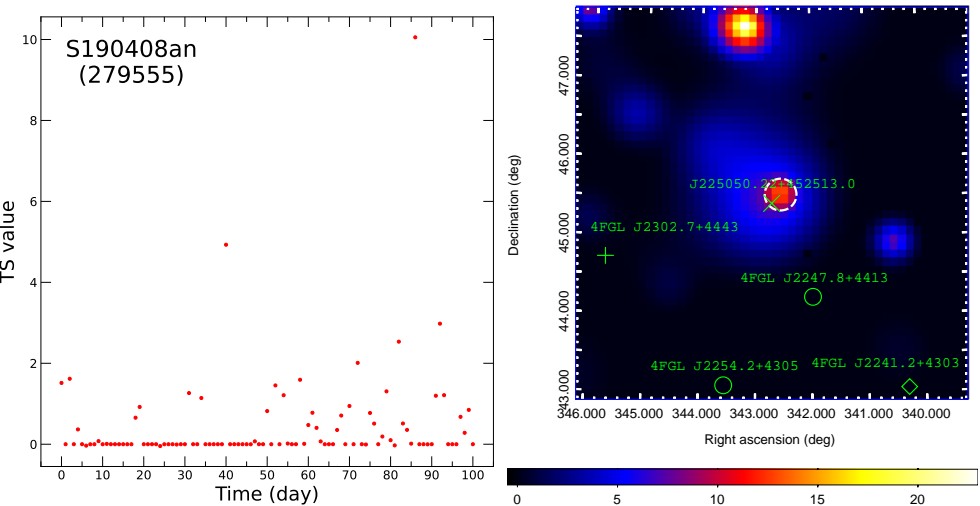

**Figure 3.** (**Left**): The 1-day binned TS curve from −1 to 100 days of the BBH merger trigger S190408an obtained for J225050.22+452513.0 (279555). A TS ≃ 10 event is seen at 86 days. (**Right**) The 0.3–500 GeV TS map with a size of 5° × 5° made for the high-TS data point on the left. The 4FGL sources in the field are marked with circles, a diamond and a plus sign, the latter two indicating a Log-Parabola and a PLSuperExpCutoff spectral form, respectively, for the two sources. The position of the galaxy target is marked with a cross, while a dashed white circle is the 2-$\sigma$ error circle (0.2 deg) determined for the candidate $\gamma$-ray source.

## 5. Discussion

Using the real-time reports on the high-probability BBH merger events detected in the LIGO/Virgo O3 run, we have carried out searches for high-energy emissions from these events. Because of the large credible areas of the merger events and the relatively low angular resolution of the *Fermi* LAT observational data we used, only seven of them have been selected to be searched. We have found 127 galaxy targets in 2MPZ that match the credible areas and the distance ranges of the seven events. For them, we have constructed different short time-bin light curves from the *Fermi* LAT data to find candidate short flaring events in the time duration of $-1$ to $100$ days of each of the merger events. Then, the TS maps were obtained for candidate flares to exclude spurious ones caused by the contamination of nearby sources. In the end, three candidates in the fields of two merger events, S200311bg and S190408an, have been found.

However, the candidates did not have sufficiently high significance, and two of them only showed in 1-day binned data. If we consider a 102 trial number (from the analysis of 1-day binned data) for each galaxy target and 127 targets, the total trial number is $\sim 10^4$, and the $\sim 3\sigma$ detection would be lowered to no significance at all. Thus no detections can be claimed. In one verification study for the purpose of determining the statistic of detecting short-duration $\gamma$-ray flares from a random position [37], 18 TS $\geq 25$ data points were found in 3-day binned light curves of 300 random positions, with each light curve consisting of 1410 data points. However, after checking the TS maps, none of them was proven to be real. Then there were 1489 data points with $9 \leq$ TS$< 25$, which unfortunately were too many to be verified with TS maps. The number indicated a probability of 3.5% for possible detections (i.e., $9 \leq$ TS $< 25$), while it is much higher than the 0.02% we have obtained here. Based on that study's results, TS $\geq 25$ is required in order to establish a certain candidate detection case.

In any case, we note that J000607.47$-$103032.8 could be an interesting case since the candidate flaring event showed in three different time-bin data at the same time. The galaxy was detected by the NRAO VLA Sky Survey (NVSS; [38]), indicating that it would likely be a radio galaxy. Radio galaxies are known to be able to emit $\gamma$-rays, as thus far, 45 identified or associated radio galaxies have been listed in 4FGL [34]. Among them, there are several (e.g., 3C 17 and PKS 2324$-$02) with redshifts comparable to J000607.47$-$103032.8. Further, detailed studies have shown that short $\gamma$-ray flares could be produced from them (e.g., [39]). In our galaxy, short $\gamma$-ray flares have been detected from micro-quasars and are thought to be related to activities of radio jets in binary systems (e.g., [37]). It is thus possible to detect short flares from such a radio galaxy, which would otherwise be non-detectable due to a relatively low luminosity. Further studies of the source and other similar (candidate) radio galaxies would help verify this case.

For searches for high-energy EM counterparts to the GW events, such as H.E.S.S. ([30]) and AGILE ([29]), different strategies have been applied, which have considered the targets and technical capability details of a facility. They were more focused on the detection of GRB-like events, although it is not clear whether such events would arise in association with BBH mergers. We have, to some degree, extended the searches to longer durations around the selected BBH merger events by taking advantage of the quick all-sky survey capability of *Fermi* LAT. While no signals were found in our searches, the work has paved the way for near-future studies. We have learnt the basic statistics on the short-flaring events we may expect to find, helping establish the criterion for true events in future searches. In the O4 run that likely starts from 2023 March, the Japanese KAGRA detector is expected to join (although in short time periods of the O4 run), which will help detect significantly more GW events with improved 90% credible areas ($\sim 30$–$50\,\mathrm{deg}^2$). Given the experience with this work, we will adjust our search strategy and hope to work on searches in real time for the whole O4 run.

**Author Contributions:** Methodology, Z.W.; formal analysis, C.R.; writing—original draft preparation, C.R.; writing—review and editing, Z.W.; supervision, Z.W. All authors have read and agreed to the published version of the manuscript.

**Funding:** We thank the referees for helpful comments. This work is supported by the Basic Research Program of Yunnan Province No. 202201AS070005 and the National Natural Science Foundation of China (12273033). C.R. thanks the Basic Research Program of the Education Division of Yunnan Province No. 2022Y055, and Z.W. acknowledges the support by the Original Innovation Program of the Chinese Academy of Sciences (E085021002).

**Data Availability Statement:** No new data were created or analyzed in this study. Data sharing is not applicable to this article.

**Conflicts of Interest:** The authors declare no conflict of interest.

## Notes

1   https://gracedb.ligo.org/superevents/public/O3/ (accessed on 9 September 2022)
2   https://www.slac.stanford.edu/exp/glast/groups/canda/lat_Performance.htm (accessed on 29 September 2022)

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
