# Peer review of "Searching for γ-ray Emission from Binary Black-Hole Mergers Detected in LIGO/Virgo O3 Run"

_universe, doi:10.3390/universe8100517_

Round 1
Reviewer 1 Report
The idea of searching for counterparts of GWs using existing catalogues is sound and is worth publishing. However, more details should be given to justify the time window used. Why the authors have decided to use -1 to 100 days ?
Please note that LAT resolution is much better than the several defress quoted on line 104 (here a reference should be added)
I see here and there strange sentences
line 70 I do not understand the meaning of certain time windows
line 94 for determing sources (maybe selecting)
line 98-99 with the range of what ?
Line 125 to the data of this target (maybe to all the data )
Author Response
>The idea of searching for counterparts of GWs using existing catalogues is
>sound and is worth publishing. However, more details should be given to
>justify the time window used. Why the authors have decided to use
>-1 to 100 days ?
Yes. Here we mainly consider the recently reported optical flare possibly
associated with a BBH merger event (Graham et al. 2020) or the strategy
the ZTF survey used. We revised the sentence to be more specific
(lines 118-119).
>Please note that LAT resolution is much better than the several defress
>quoted on line 104 (here a reference should be added)
It is several degrees? According to Fermi LAT performance pages, the 68%
containment angle (i.e., the angular resolution) is about 2 deg at 300 MeV
and 6 deg at 100 MeV:
https://www.slac.stanford.edu/exp/glast/groups/canda/lat_Performance.htm
We added the performance webpage as a footnote reference (page 3).
>I see here and there strange sentences
>line 70 I do not understand the meaning of certain time windows
We tried to say it is a transient event and observations in some time ranges
are required.
We changed "certain time windows" to "uncertain time ranges requiring
observational coverage"
>line 94 for determing sources (maybe selecting)
Done.
>line 98-99 with the range of what ?
Done. Changed it to "the given redshift range"
>Line 125 to the data of this target (maybe to all the data )
This sentence was not clear. We revised it. Other revisions were made at
the place (beginning of the third paragraph of Section 3) due to comments
from the other referees.
Reviewer 2 Report
This is a solid work that I believe deserves publications. The English writing is quite good for writing in English as a foreign language, but it could be improved by some editing done by a native English speaker. The following are some suggestions that might improve the manuscript (not in order of importance) and some questions that arose when reading:
1. In section 3 the first sentence notes that the LAT energy range extends down to 100 MeV, but in the next sentence events are rejected below 300 MeV. That's probably reasonable, but the reader could benefit from an explanation why. Hopefully this cut was made before looking for possible signals, not after.
2. The 15 by 15 degree ROI apparently corresponds to an inscribed circular radius of 10.6 degrees. I think LAT analyses typically include known sources to larger angles (e.g. 20 degrees for 100 MeV, but somewhat less for 300 MeV). I would worry a little bit if there were a bright source just outside this ROI, so it could be a good idea to check on that.
3. The catalog used, based on the references, appears to be 4FGL-DR3. If so, specify that as opposed to just 4FGL.
4. Were there any extended sources in the ROIs studied?
5. Did the authors check that the sun and moon did not pass through the ROI during the relevant time period?
6. What were the convergence criteria for the fits? Were problems encountered with non-convergence?
7. Regarding the first flaring event, it might be interesting for the reader to know how many events are responsible for that flare, and maybe something about the energies of those events.
8. For the galaxy of interest, there are a couple of 4FGL sources nearby, so it could be interesting to estimate the probability of the events of interest come from those sources (e.g. using gtsrcprob). The authors could also consult some LAT collaboration resources for variability in known sources, such as FAVA (https://fermi.gsfc.nasa.gov/ssc/data/access/lat/FAVA/), or the more recent Fermi LAT Light Curve repository (https://fermi.gsfc.nasa.gov/ssc/data/access/lat/LightCurveRepository/). It would be helpful to specify in the draft the identities of those known sources, so the reader doesn't have to look them up.
9. Can the caption of Figure 2 explain why one of the sources is marked with a diamond instead of circle?
10. In the trials calculation, I think that the number of trials should include more than just the number of bins. In particular, it should include the number of galaxy targets studied (i.e. times the number of bins for each). That will give a trials number that is very large compared with the 102 stated. This is my only serious criticism concerning validity of the stated results.
11. For completeness, the manuscript could mention the version of Science Tools used, as well as the IRF (instrument response function).
12. Figure 1 right: what are the other red concentrations at the right-hand side of the field? Apparently they do not correspond to known sources? Or are they just random fluctuations? Similarly for the red/yellow concentrations at the top of Figure 3 right.
13. Line 70: is the word supposed to be "uncertain" instead of "certain"?
Author Response
>This is a solid work that I believe deserves publications. The English
>writing is quite good for writing in English as a foreign language, but it
>could be improved by some editing done by a native English speaker.
>The following are some suggestions that might improve the manuscript
>(not in order of importance) and some questions that arose when reading:
>1. In section 3 the first sentence notes that the LAT energy range extends
>down to 100 MeV, but in the next sentence events are rejected below 300 MeV.
>That's probably reasonable, but the reader could benefit from an explanation
>why. Hopefully this cut was made before looking for possible signals, not
>after.
Yes. We revised the energy range of LAT to be 50 MeV -- 1 TeV, which is
formally stated in every Fermi LAT collaboration papers. We added
the reasons for choosing >0.3 GeV, that is to avoid those uncertainties
which would be caused by including <0.3 GeV events (lines 124--128).
>2. The 15 by 15 degree ROI apparently corresponds to an inscribed circular
>radius of 10.6 degrees. I think LAT analyses typically include known sources
>to larger angles (e.g. 20 degrees for 100 MeV, but somewhat less for 300 MeV).
> I would worry a little bit if there were a bright source just outside this
>ROI, so it could be a good idea to check on that.
Because we only performed unbinned analysis, the RoI actually was a circle
with a 15-deg radius. Sorry for the mistake in the text. We changed it to
be "a 15-deg radius circle" (line 128).
This work was done in the beginning of this year, and since we were invited to
submit it to this special Issue, we waited for half of a year and then wrote
this manuscript. Also there were some mis-communications between the first
and second author.
We have checked the details of our analysis. Now we believe every step is
clear.
>3. The catalog used, based on the references, appears to be 4FGL-DR3. If so,
>specify that as opposed to just 4FGL.
It was DR2, because at the time of the analysis, DR3 was not available.
We changed it to 4FGL-DR2 and the reference. Note that we have
compared the details between DR2 and DR3, and there are no significant changes
between the two versions.
>4. Were there any extended sources in the ROIs studied?
For J2250+4525 (the third flare case), there are two diffuse sources in the RoI, but they are 8.9 and 13.6 deg away. For the other two cases,
there are no extended sources in the RoIs. We added sentences mentioning
this (lines 163--171 in the first paragraph of Section 4).
>5. Did the authors check that the sun and moon did not pass through the ROI
>during the relevant time period?
No. Now since this is raised, we have checked the positions of the Sun and
Moon, and they were not close to the RoIs. We added sentences mentioning
this in the text. See the above reply to Point 4.
>6. What were the convergence criteria for the fits? Were problems encountered
> with non-convergence?
We checked the regular running result information for the convergence. We did
meet non-convergence cases, for which we tried either re-running the analysis
or changing the parameter ranges. If still could not pass, as we mostly ran
on zero-signal cases, we assigned it as a TS=0 case.
We added a sentence at the end of paragraph 3 in Section 3 (lines 149--151).
>7. Regarding the first flaring event, it might be interesting for the reader
>to know how many events are responsible for that flare, and maybe something >about the energies of those events.
Yes. We checked the events and found that the primary one was a 11.4 GeV photon.
We added a paragraph at the end of Section 4.1.
>8. For the galaxy of interest, there are a couple of 4FGL sources nearby,
>so it could be interesting to estimate the probability of the events of
>interest come from those sources (e.g. using gtsrcprob). The authors could
>also consult some LAT collaboration resources for variability in known
>sources, such as FAVA (https://fermi.gsfc.nasa.gov/ssc/data/access/lat/FAVA/),
>or the more recent Fermi LAT Light Curve repository
>(https://fermi.gsfc.nasa.gov/ssc/data/access/lat/LightCurveRepository/).
>It would be helpful to specify in the draft the identities of those known
>sources, so the reader doesn't have to look them up.
Yes. We checked the probabilities as well as the variabilities of nearby
sources shown in the TS maps. There were two sources with high
Variability_Index values. However none of them plus other nearby sources
likely affected our results, for which we also checked the TS maps that
contain the nearby sources (i.e., they are kept in the TS maps).
We added sentences about this. See the reply above for Point 7.
>9. Can the caption of Figure 2 explain why one of the sources is marked with
>a diamond instead of circle?
Yes. We added the descriptions for the diamond and plus-sign symbols
in all the TS map captions. They are used in the catalog to indicate
different spectral forms for the sources.
>10. In the trials calculation, I think that the number of trials should
>include more than just the number of bins. In particular, it should include
>the number of galaxy targets studied (i.e. times the number of bins for each).
>That will give a trials number that is very large compared with the 102 stated.
>This is my only serious criticism concerning validity of the stated results.
Yes. As there is another comment about this, we revised the discussion there
(second paragraph of Section 5).
>11. For completeness, the manuscript could mention the version of Science
>Tools used, as well as the IRF (instrument response function).
Yes. We added the versions of the Fermitools and IRF. We also added the tools
used in the analysis.
>12. Figure 1 right: what are the other red concentrations at the right-hand
>side of the field? Apparently they do not correspond to known sources? Or
>are they just random fluctuations? Similarly for the red/yellow concentrations
>at the top of Figure 3 right.
Yes. We met many such cases when analyzing LAT data in our work, but for a
fraction of them if the cases were sufficiently bright ones, our investigations
for their nature often resulted nothing with certainty.
>13. Line 70: is the word supposed to be "uncertain" instead of "certain"?
Done. We also added a few words at the end as requested by another referee.
Reviewer 3 Report
This an analysis framework for gamma-ray data coincident with a gravitational wave event, using an optical catalogue of galaxies to fix the center of the search region. The context of the special issue and the content of the manuscript seem proper to publication. Nevertheless I have several general comments that I believe need addressing.
- English level could be improved with the help of the editors.
- While the proposed analysis seem reasonable, the introduction does not really discuss in what manner it differs from other techniques published in the past. As a result is is impossible to properly evaluate the originality of the present work.
- The authors use the LAT data but make no mention of the GBM data. It should be discussed.
- LAT data could be gathered during a TOO triggered by the GW. The authors need to provide this information for their targets.
- The authors mention matching optical photometric redshift in the 2MASS catalogue with the GW redshift estimate, without being explicit about what they did exactly: how are the uncertainties taken into account, etc... More generally, the draft has too many occurrences of qualitative assertions not backed by quantitative or more explicit statements. For instance on l.105, the criterion to exclude some merger events is not specified. Even more serious, on l.135 it is absolutely not clear what criterion is used to associate a gamma-ray candidate to a 2MASS location, nor to discard possible contamination from 4FGL nearby sources (l.162 for instance where no quantitative statement is provided to the claim that there is no contamination).
- in the TS maps, the 4FGL circles should reflect the size of the position uncertainty on a 4FGL source
- The TS threshold is very low (and by the way TS>9 does not per se mean >3sigma detection), which I can understand given the exploratory nature of the work. Nevertheless, there is a simple way to assess proper TS, or at least provide the equivalent of a p-value : just apply the analysis to random times outside of the considered interval of analysis. As the claimed goal of the work is to prepare for future GW observations, this seems like an important stage to validate in such an exploratory work.
- Careful on l.152 : the fact that an excess may look as a point source is irrelevant to the plausibility of the detection, as the TS map by construction forces fits of the amplitude of a point source in a grid of positions.
- I believe the LAT is scanning the whole sky in 90', not 3 hours, but this needs a cross check.
- In the conclusions, while Kagra is a welcomed addition to the network, during O4 its usefulness will be limited to a small volume.
Author Response
>This an analysis framework for gamma-ray data coincident with a gravitational
>wave event, using an optical catalogue of galaxies to fix the center of
>the search region. The context of the special issue and the content of
>the manuscript seem proper to publication. Nevertheless I have several
>general comments that I believe need addressing.
>- English level could be improved with the help of the editors.
>- While the proposed analysis seem reasonable, the introduction does not
>really discuss in what manner it differs from other techniques published
>in the past. As a result is is impossible to properly evaluate
>the originality of the present work.
Good point! We added two sentence in the last paragraph of Introduction.
>- The authors use the LAT data but make no mention of the GBM data. It
>should be discussed.
Yes. See above.
>- LAT data could be gathered during a TOO triggered by the GW. The authors
>need to provide this information for their targets.
Since 2018 March, no ToO observations are performed.
We added a sentence at the end of the first paragraph of Section 3.
>- The authors mention matching optical photometric redshift in the 2MASS
>catalogue with the GW redshift estimate, without being explicit about what
>they did exactly: how are the uncertainties taken into account, etc... More
>generally, the draft has too many occurrences of qualitative assertions
>not backed by quantitative or more explicit statements. For instance
>on l.105, the criterion to exclude some merger events is not specified.
>Even more serious, on l.135 it is absolutely not clear what criterion is
>used to associate a gamma-ray candidate to a 2MASS location, nor to discard
>possible contamination from 4FGL nearby sources (l.162 for instance where no
>quantitative statement is provided to the claim that there is no
>contamination).
Yes.
We added a sentence about the redshift range and accuracy of 2MPZ
(lines 105--106).
We added the information for five excluded merger events, four of them
being too-dense cases and one being zero match case. The added text is at
the beginning of the third paragraph of Section 2.
We added the information for the non-detection and detection cases in
paragraph 4 of Section 3.
We derived the error circles for the reported three candidate flaring sources.
The text added about this is marked in boldface in the Results Section
(the places describing each candidate detection). We also revised Figures 1--3
by adding the error circles.
>- in the TS maps, the 4FGL circles should reflect the size of the position
>uncertainty on a 4FGL source
Regularly we have been showing positions without uncertainties for many
results; otherwise it would be a little bit complicated. Also note that
our targets are mostly away from the Galactic plane, and so those in the fields
should be point sources with <0.1 deg positional uncertainties and
not some extended ones with large regions to be marked (for Galactic cases
such as SNRs, specific marks are needed).
However, the comment reminds us that we should mark the error circles for
our detected candidate gamma-ray sources. See the reply to the comment right
above.
>- The TS threshold is very low (and by the way TS>9 does not per se mean
>>3sigma detection), which I can understand given the exploratory nature
>of the work. Nevertheless, there is a simple way to assess proper TS, or
>at least provide the equivalent of a p-value : just apply the analysis to
>random times outside of the considered interval of analysis. As the claimed
>goal of the work is to prepare for future GW observations, this seems like
>an important stage to validate in such an exploratory work.
Yes. We (author Z. Wang) have done a study before (although not formally
published, but part of the results were described in Xing & Wang 2021), which
indicates that TS>=25 is required for likely detection of a short flaring
event. We added the results to the discussion in the second paragraph of
Discussion.
>- Careful on l.152 : the fact that an excess may look as a point source is
>irrelevant to the plausibility of the detection, as the TS map by
>construction forces fits of the amplitude of a point source in a grid of
>positions.
Yes. Sorry for ambiguousness here. It is more about comparing to the situations
that high TS values are detected at some positions but found out due to some
extended structures in TS maps (easily happen in the Galactic plane regions).
We revised the sentence (now lines 180--181).
>- I believe the LAT is scanning the whole sky in 90', not 3 hours, but this
>needs a cross check.
It is "covering the whole sky every three hours". This information can be
found at Fermi LAT webpages.
>- In the conclusions, while Kagra is a welcomed addition to the network,
>during O4 its usefulness will be limited to a small volume.
Yes. We know about this. We added a few words at the end of the last
paragraph pointing this out.
Reviewer 4 Report
The author explores the emission of gamma rays sourced by BBH using the data from GraceDB, LAT, Fermi, and 2MPZ. The nature of these events and their statistical significance are studied in great detail. All in all, I think the article should be published in its current state.
Author Response
>The author explores the emission of gamma rays sourced by BBH using the data
>from GraceDB, LAT, Fermi, and 2MPZ. The nature of these events and their
>statistical significance are studied in great detail. All in all, I think
>the article should be published in its current state.
Thanks for the comments!